# The Role of Self-Efficacy, Motivation, and Connectedness in Dropout Intention in a Sample of Italian College Students

Chiara Buizza [1,*], Herald Cela [1,2], Giulio Sbravati [1], Sara Bornatici [1], Giuseppe Rainieri [1]
and Alberto Ghilardi [1]

1    Department of Clinical and Experimental Sciences, University of Brescia, Viale Europa 11, 25123 Brescia, Italy;
     herald.cela@uni-graz.at (H.C.); giulio.sbravati@unibs.it (G.S.); sara.bornatici@unibs.it (S.B.);
     giuseppe.rainieri@unibs.it (G.R.); alberto.ghilardi@unibs.it (A.G.)
2    Institute of Psychology, University of Graz, Universitätsplatz 2, 8010 Graz, Austria
*    Correspondence: chiara.buizza@unibs.it

**Abstract:** Dropout is a critical concern in higher education, with a considerable number of students leaving within the first two years of university. Dropout affects students' well-being and their academic and career prospects, and institutions' retention and graduation rates. The aim of this study was to explore the mediating role of motivation and cognitive strategies for learning in the relationship among self-efficacy, connectedness, and university dropout intention. A total of 790 Italian college freshmen were involved in this study. The sample was recruited through a web survey consisting of the Academic Motivation Scale, Perceived School Self-Efficacy Scale, University Connectedness Scale, and Self-Regulated Knowledge Scale-University. The freshmen's intentions to drop out were assessed with five questions. The average age of the freshmen was 20.9 years, most of them were female, and were attending a degree program in the medical area. The results show that self-efficacy is the most important predictor of dropout intentions, followed by university connectedness. Self-regulated knowledge has an important role in predicting dropout intention by acting as a mediator between self-efficacy and motivation.This study underlines the importance of investing in training and orientation interventions in order to develop the skills to face the university path, increasing self-efficacy, motivation, and consequently students' well-being.

**Keywords:** freshmen; dropout; cognitive strategies; self-efficacy; motivation; connectedness

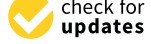



## 1. Introduction

Dropout is a critical concern in higher education, with a considerable number of students leaving within the first two years of university [1]. Dropout not only involves the personal sphere of the students in terms of compromising their well-being and prospects for the future but also has significant implications from a social and economic point of view. The literature shows that dropout is a complex phenomenon that includes many factors. Several studies highlighted the role of self-efficacy, motivation, and connectedness [2–5], with self-efficacy and connectedness being particularly important in predicting academic retention [6].

Self-efficacy refers to the belief in one's ability to perform a task successfully. Research suggests that higher levels of self-efficacy are associated with increased academic achievement and persistence [7,8]. Motivation, on the other hand, is the driving force that initiates, directs, and sustains behavior toward a particular goal. Seminal studies already pointed out how students who are more intrinsically motivated tend to experience higher academic achievements and persistence [8]. This motivation is that to engage in a behavior because of the inherent satisfaction of the activity rather than the desire for a reward or specific outcome [9]. On the contrary, students who are extrinsically motivated are more likely to drop out [10], because their motivation arises from external factors. Considered together,

self-efficacy and intrinsic motivation are positively associated with academic persistence, while extrinsic motivation is negatively associated [11].

Connectedness, on the other hand, reflects the extent to which students feel connected to their peers, professors, and the institution as a whole. Often defined as a sense of belonging and social support within academic settings [12,13], it has been found to influence both academic achievements and retention in a positive way [14,15].

Students' ability to control their own learning process may also be another important factor in academic retention and persistence [16,17]. Self-regulated learning is the process by which learners take control of their own learning by setting goals, monitoring their progress, and adapting their learning strategies accordingly [18,19]. Several studies showed that self-regulated learning is a crucial factor in predicting academic success and persistence, as well as reducing the likelihood of dropout [16]. Students who engage in self-regulated learning are more likely to persist in their studies and achieve better grades, as they are able to regulate their motivation, cognition, and behavior toward learning [17].

Given the importance of academic retention and the role of self-efficacy, motivation, and connectedness, it is crucial to investigate their complex interactions in order to develop effective interventions to promote student success. While previous studies examined the individual relationships among all these factors, less is known about how they interact in explaining the intention to drop out of university, and even less is known on the potential co-effect introduced by self-regulated learning.

In Italy, it is estimated that dropout rates are particularly higher between the first and second year of university studies [20,21]. This is particularly alarming if we consider that Italy has the lowest percentage of graduates in Europe [22]. The dropping out of higher education has implications for both students and universities, and, therefore, identifying the variables associated with dropout makes it possible to develop actions aimed at reducing its occurrence.

The aim of this study is to explore the mediating role of motivation and of cognitive strategies for learning in the relationships that link self-efficacy and connectedness to dropout intention in a sample of college freshmen. By exploring the relationships among these factors, this study seeks to provide a better understanding of the complex mechanisms underlying academic persistence. In testing these relationships, a wide range of potential predictors will be considered, including demographic variables, socio-economic status, and academic background.

## 2. Materials and Methods

This work is a cross-sectional study involving freshmen from a university in Northern Italy. This study was conducted in June 2022. Freshmen agreed to take part in this study after reading an information note and by signing an informed consent form. They were informed that the participation was voluntary and that the survey was anonymous. The web survey was created with LimeSurvey (www.limesurvey.org, accessed on 1 June 2022), a proprietary survey tool that allows completely anonymous data collection. The software automatically sends a personal link to the survey via email to each participant. Once a participant completes the survey, LimeSurvey removes any participant identifiers from the survey data. The survey was implemented following the guidelines proposed by Pealer and Weiler [23].

### 2.1. Measures

For the purposes of this study, the following scales were used:

The adapted form of the Academic Motivation Scale (AMS), developed by Biasi et al. [24]. It is a questionnaire developed on the basis of Self-Determination Theory [10]. The AMS is composed of 20 items rated on an 11-point Likert scale ranging from 0 (Not at all true) to 10 (Completely true). The items are grouped into 5 sub-scales: lack of motivation, external regulation, introjected regulation, identified regulation, and intrinsic regulation. The total score ranges from 0 to 40, where a higher score corresponds to a greater adherence

to the construct of the single sub-scale. The Italian version showed good psychometric properties with Cronbach's alpha values ranging from 0.73 to 0.91 [25].

The reduced and adapted form of the Perceived School Self-Efficacy Scale (PSSES), developed by Biasi et al. [24]. It is a tool to assess the perception that students have of their ability to regulate and focus on the studying process [26]. The PSSES is composed of 9 items rated on a 5-point Likert scale ranging from 1 (Not capable at all) to 5 (Fully capable). The total score ranges from 9 to 45, where a higher score corresponds to a higher level of self-efficacy perceived by the student. The Italian version proved to be a reliable instrument with Cronbach's alpha values ranging from 0.83 to 0.87 [26].

The University Connectedness Scale (UCS), which assesses the degree of support and membership perceived by students with respect to their university [27]. The UCS consists of 18 items rated on a 7-point Likert scale, ranging from 1 (Not at all) to 7 (All the time). The total score ranges from 18 to 126, where the higher the score, the higher the students' perception of belonging and support within their university. The UCS has a strong internal consistency with a Cronbach's alpha of 0.88 [28].

The Self-Regulated Knowledge Scale-University (SRKS-U) developed on the basis of Pintrich's Theory of Self-regulated Knowledge [16]. It assesses the frequency with which students implement different cognitive strategies. The SRKS-U consists of 15 items rated on a 5-point Likert scale ranging from 1 (Never) to 5 (Always or nearly always). The SRKS-U is composed of 5 sub-scales evaluating the use of predefined cognitive processes: knowledge extraction, knowledge networking, knowledge practice, knowledge critique, and knowledge monitoring. The score of each sub-scale ranges from 3 to 15, where a higher score corresponds to a greater use of that cognitive strategy. The Italian version proved to be a reliable instrument with Cronbach's alpha values ranging from 0.70 to 0.80 [29].

The freshmen's intentions to drop out were assessed with 5 questions resulting from the Hardre and Reeve scale [30], adapted in the study by Biasi et al. [24]. The 5 questions were (1) How often do you think about dropping out of university and doing something else? (2) How often do you feel insecure about continuing your university studies year after year? (3) How often do you consider not continuing your university studies? (4) How often do you think to drop out of university? (5) How often do you think to drop out of your program to take up another one? The answer choices for each item are based on a 5-point Likert scale ranging from 1 (Never) to 5 (Always).

Freshmen also filled in an assessment form with information regarding their socio-demographic and academic features.

*2.2. Statistical Analysis*

This study employed SPSS 29.0 and AMOS 28 to conduct descriptive statistics, multiple regression analysis, Confirmatory Factor Analysis (CFA), and Structural Equation Modeling (SEM).

Descriptive statistics were performed to summarize the characteristics of the sample and the study variables of interest. Outliers and missing data were also examined and handled appropriately, with cases that had missing data for entire scales being excluded from the analysis, whereas missing cases for singular items were filled following the multiple imputation strategy [31]. Data screening was performed through case-by-case analysis to distinguish unengaged respondents ("yea-sayers"). In most of the target scales, observations with low variability (SD < 0.25) were excluded from the analysis.

An explorative analysis by means of hierarchical regression was performed in order to identify significant predictors for the dropout outcome. All socio-demographic variables were considered as potential predictors. The predictors entered in the first block were chosen based on their consistent associations with academic performance and persistence in prior research [32–34]. Gender [32], parents' education [35], field of study [33], the proportion of class attended [32], and socio-economic conditions [36,37] have all been shown to be significant predictors of academic performance and persistence. In the second block,

we entered the remaining variables, including living area, relationship status, working status, and incoming school.

CFA was conducted to assess the measurement model fit of the six latent constructs: self-efficacy, connectedness, self-regulated knowledge, intrinsic motivation, extrinsic motivation, and dropout. For self-efficacy, connectedness, self-regulated knowledge, and dropout, all the observed indicators contained in the reported scales were included. Concerning motivation, sub-scales of the AMS scale were re-arranged following the Self-Determination Theory continuum [9]. Therefore, to represent the construct of intrinsic motivation items from the sub-scales, intrinsic regulation and identified regulation were included. To represent the construct of extrinsic motivation items from the sub-scales, external regulation and introjected regulation were included. To evaluate the goodness-of-fit of the model, the following fit indices were used: Chi-square test, Comparative Fit Index (CFI), Goodness-of-Fit Index (GFI), Tucker–Lewis Index (TLI), Root Mean Square Error of Approximation (RMSEA), and Standardized Root Mean Square Residual (SRMR). Acceptable threshold values for the fit indices were CFI and TLI > 0.90, GFI > 0.90, RMSEA < 0.08, and SRMR < 0.08 [38]. To improve the model fit, modification indices were examined and covariances were created between residuals within factors when the expected change in the Chi-square test was of at least 20 points. Item reliability was assessed by examining the standardized regression weights, and items with a loading less than 0.50 were deleted [39]. The adoption of standardized regression weights was preferred to ensure comparability of variables and address the issue of different scaling systems. Internal consistency for the measures was assessed using Cronbach's alpha, adopting a threshold of 0.70 to consider the scale reliable [39]. Convergent validity was assessed by examining the Average Variance Extracted (AVE) for each construct, with values above 0.50 indicating acceptable convergent validity [40]. Discriminant validity was assessed through the Heterotrait–Monotrait (HTMT) ratio [41].

SEM was performed to examine the relationships between self-efficacy and connectedness on dropout, hypothesizing a mediational effect for intrinsic, extrinsic motivation, and self-regulated knowledge. The hypothesized model was specified using the latent variables and observed indicators from the CFA, and then estimated with the maximum likelihood method. Multi-group analysis was conducted to examine the differences in the model across groups defined by predictors identified by hierarchical regression at the previous step. Model comparison was conducted using nested models, where a more restrictive model was compared to a less restrictive model using the Chi-square difference test. The invariance of the model across groups was examined through the configural and metric invariance [42]. Mediation analysis was conducted in the SEM to test for total, direct, and indirect effects of the hypothesized model. All statistical analyses were conducted using an Alpha of $p < 0.05$.

## 3. Results

### 3.1. Characteristics of the Sample

A total of 790 college freshmen were involved in this study. The average age was 20.9 years (SD = 4.2), and most were female (63.4%). The majority of the sample (94.2%) was of Italian nationality. The majority of the sample attended a degree program in the medical area (42.7%), followed by engineering (25.9%), economics (24.7%), and law (6.7%). The average GPA was 24.7 (SD = 3.1) out of 30. The majority (81.3%) attended class with a frequency greater than 70%. Table 1 shows further features of the sample.

Concerning the dropout intentions, 14.5% of the sample thought of dropping out of university and doing something else, 12.7% considered not continuing their university studies, 18.3% felt insecure about continuing their university studies year after year, 12.1% thought of dropping out of their program to take up another, and 11.0% intended to drop out of university.

**Table 1.** Characteristics of the sample (n = 790).

| Variables | n (%) |
|:---:|:---:|
| *Gender* | |
| Male | 284 (35.9) |
| Female | 501 (63.4) |
| Other | 5 (0.6) |
| *Relationship status* | |
| Single | 451 (57.1) |
| Married/cohabitant | 25 (3.2) |
| In a relationship | 311 (39.4) |
| Separated/divorced/widower | 3 (0.4) |
| *Incoming school* | |
| High school | 437 (55.3) |
| Technical school | 290 (36.7) |
| Vocational school | 63 (8.0) |
| *Mother's educational level* | |
| Primary school | 22 (2.8) |
| Secondary school | 253 (32.0) |
| High school | 351 (44.4) |
| University/postgraduate specialization | 164 (20.7) |
| *Father's educational level* | |
| Primary school | 28 (3.5) |
| Secondary school | 276 (34.9) |
| High school | 346 (43.8) |
| University/postgraduate specialization | 140 (17.7) |
| *Area of residence* | |
| Rural (Up to 100.000 residents) | 613 (77.6) |
| Urban (Over 100.000 residents) | 177 (22.4) |
| *Income bracket* | |
| Up to 36.151.98 € | 427 (54.1) |
| From 36.151.99 to 70.000 € | 232 (29.4) |
| From 70.001 to 100.000 € | 90 (11.4) |
| Over 100.000 € | 41 (5.2) |
| *Working student* | |
| Yes | 267 (33.8) |
| No | 523 (66.2) |

*3.2. Multiple Regression*

The final regression model explained a substantial proportion of the variance in academic dropout, with an adjusted $R^2$ of 0.84. The omnibus test of the final model was highly significant, $F_{(4.780)} = 631.06$, $p < 0.001$. The model included six predictors: gender ($\beta = 0.26$, $p < 0.001$), father's education level ($\beta = 0.20$, $p < 0.001$), proportion of class attendance ($\beta = -0.14$, $p = 0.027$), area of study ($\beta = 0.13$, $p < 0.001$), relationship status ($\beta = 0.08$, $p = 0.006$), and incoming school ($\beta = 0.26$, $p < 0.001$).

*3.3. Confirmatory Factor Analysis (CFA)*

All the fit measures defined and used to assess the model were above their respective commonly accepted thresholds, expressing an overall goodness-of-fit of the final measurement model: $\chi^2_{(600)} = 1469.69$, $p < 0.001$, CFI = 0.96, TLI = 0.95, RMSEA = 0.04, and SRMR = 0.06 ($\chi^2/df = 2.45$). Factor loadings for all items included in the final version of the model, along with Cronbach's alpha, Composite Reliability (CR), and Average Variance Explained (AVE), are presented in Table 2. Cronbach's alpha for each construct in this study was found to be above the required level of 0.70, ranging from 0.87 to 0.94. Composite Reliability values ranged from 0.86 to 0.93, above the 0.70 benchmark. Therefore, internal consistency was established for each construct in this study. The AVE values were above the threshold value of 0.50; thus, convergent validity was validated. Finally, discriminant

validity was assessed using the HTMT ratio, with all the values being less than the required limit of 0.85. Hence, discriminant validity was established.

**Table 2.** Standardized regression weights for all items.

| Scale/Item | Estimate | $\alpha$ | CR | AVE |
|---|---|---|---|---|
| Extrinsic motivation | | 0.870 | 0.871 | 0.494 |
| AMS_1 | 0.823 | | | |
| AMS_6 | 0.931 | | | |
| AMS_11 | 0.876 | | | |
| AMS_13 | 0.775 | | | |
| Intrinsic motivation | 8/8 | 0.938 | 0.925 | 0.611 |
| AMS_4 | 0.892 | | | |
| AMS_5 | 0.718 | | | |
| AMS_9 | 0.813 | | | |
| AMS_10 | 0.700 | | | |
| AMS_14 | 0.889 | | | |
| AMS_15 | 0.680 | | | |
| AMS_19 | 0.908 | | | |
| AMS_20 | 0.593 | | | |
| Connectedness | | 0.868 | 0.865 | 0.479 |
| UCS_3 | 0.597 | | | |
| UCS_5 | 0.671 | | | |
| UCS_8 | 0.771 | | | |
| UCS_10 | 0.734 | | | |
| UCS_12 | 0.591 | | | |
| UCS_13 | 0.741 | | | |
| UCS_14 | 0.719 | | | |
| Self-efficacy | | 0.875 | 0.871 | 0.495 |
| SASP_1 | 0.790 | | | |
| SASP_2 | 0.794 | | | |
| SASP_3 | 0.705 | | | |
| SASP_5 | 0.731 | | | |
| SASP_6 | 0.717 | | | |
| SASP_7 | 0.583 | | | |
| SASP_8 | 0.573 | | | |
| Dropout | | 0.930 | 0.927 | 0.761 |
| dropout_1 | 0.838 | | | |
| dropout_2 | 0.771 | | | |
| dropout_3 | 0.936 | | | |
| dropout_4 | 0.934 | | | |
| Self-learning | | 0.870 | 0.871 | 0.494 |
| SRK_6 | 0.565 | | | |
| SRK_8 | 0.762 | | | |
| SRK_3 | 0.646 | | | |
| SRK_13 | 0.677 | | | |
| SRK_15 | 0.728 | | | |
| SRK_10 | 0.782 | | | |
| SRK_5 | 0.734 | | | |

*Note.* CR = Composite Reliability; AVE = Average Variance Explained; $\alpha$ = Cronbach's alpha. Items are presented as retained in the CFA over the total number of items on the original scale.

Measurement invariance was performed on the final measurement model to conduct robust multi-group analysis on the selected predictors. Fit indices for every factor, including configural invariance and metric invariance, are reported in Table 3. In our model, configural invariance was achieved, observing that the fit indices are above the cut-off values when analyzing the measurement model separately for groups. To achieve metric invariance, the constrained model should not be worse than the unconstrained one at the configural level observed in the differences-of-fit indices. When analyzing the change in models considering the difference in Chi-square, there were issues for metric invariance

concerning parents' education, class attendance, and field of study. However, given that the change in the other fit indices is minimal (<0.001) and in line with the literature suggesting considering multiple fit indices [43,44], we decided to keep all the factors for the multi-group analysis.

**Table 3.** Measurement invariance.

| | **Configural Invariance** | | | | | **Metric Invariance** | | | | |
|---|---|---|---|---|---|---|---|---|---|---|
| **Factor** | $\chi^2$ **(df)** | *p* | **CFI** | **TLI** | **RMSEA** | $\Delta\chi^2$ **(df)** | *p* | **ΔCFI** | **ΔTLI** | **ΔRMSEA** |
| Gender | 2227.59 (1200) | <0.001 | 0.948 | 0.942 | 0.033 | 40.07 (31) | 0.127 | 0.000 | 0.001 | 0.000 |
| Parents' educational level * | 3066.13 (1800) | <0.001 | 0.934 | 0.927 | 0.031 | 99.88 (62) | 0.002 | 0.002 | 0.000 | 0.031 |
| Attendance | 2177.62 (1200) | <0.001 | 0.950 | 0.945 | 0.032 | 57.14 (31) | 0.003 | 0.001 | 0.000 | 0.032 |
| School | 3215.44 (1800) | <0.001 | 0.931 | 0.923 | 0.032 | 70.83 (62) | 0.207 | 0.001 | 0.002 | 0.001 |
| Study | 3938.97 (2400) | <0.001 | 0.925 | 0.917 | 0.029 | 157.55 (93) | <0.001 | 0.003 | 0.001 | 0.000 |
| Relationship ** | 2239.61 (1200) | <0.001 | 0.947 | 0.941 | 0.034 | 43.02 (31) | 0.074 | 0.001 | 0.001 | 0.000 |

*Note*. * = For the factor of parents' educational level, the only groups compared were secondary school, high school, and university. ** = For the factor of relationship, the only groups compared were single, engaged.

*3.4. Structural Equation Modeling (SEM)*

After assessing the goodness-of-fit of the measurement model, the full SEM was constructed, as depicted in Figure 1, considering the associations between constructs reported in the research literature.

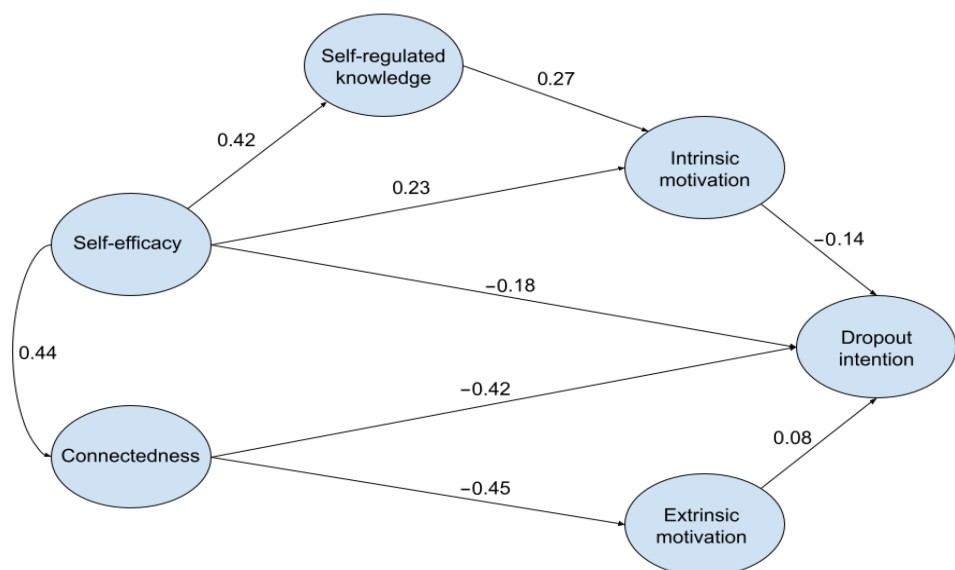

**Figure 1.** Full Structural Equation Model. *Note*. Path coefficients expressed as standardized regression weights.

All fit measures used to evaluate the model were above their respective commonly accepted thresholds, indicating an overall good fit of the final structural model: $\chi^2$ (600) = 1.594, $p < 0.001$, CFI = 0.95, TLI = 0.95, and RMSEA = 0.04 ($\chi^2$/df = 2.63). The model includes the exogenous variables' connectedness (CO) and self-efficacy (SE), mediators' motivation (IM, EM) and self-regulated knowledge (SL), and dropout intention (DO) as the final outcome. All paths in the model were found to be significant at the 0.01 level, except for the path connecting extrinsic motivation (EM) to dropout (DO). Therefore, the results of

the model suggest that self-efficacy has a partially mediated effect on dropout through self-regulated knowledge and intrinsic motivation, in addition to the direct effect of connectedness. To gain a more comprehensive understanding of the relationships in the model, serial mediation analysis was conducted. Although only indirect effects are presented here, Table 4 displays the full results of the mediation analysis, including the direct, indirect, and total effects. Partial mediation by SL was found for both the SE-IM path, β = 0.11, 95% CI [0.08, 0.16], and the SE-DO path, β = −0.03, 95% CI [−0.05, −0.01]. Meanwhile, IM partially mediated the SL-DO effect, β = −0.04, 95% CI [−0.07, −0.02]. Finally, partial mediation was confirmed for the joint effect of both mediators (SL and IM) in the path connecting SE to the final outcome (DO), β = −0.03, 95% CI [−0.05, −0.01].

**Table 4.** Total, direct, and indirect effects.

| Path | Total Effects | | Direct Effects | | Indirect Effects | | Interpretation |
|---|---|---|---|---|---|---|---|
| | β | 95% CI | β | 95% CI | β | 95% CI | Interpretation |
| SE-SL | 0.42 | [0.35, 0.49] | 0.42 | [0.35, 0.49] | - | - | Direct relationship |
| SE-IM | 0.35 | [0.26, 0.42] | 0.23 | [0.15, 0.32] | 0.11 | [0.08, 0.16] | Partial mediation |
| SE-DO [Ind1] | −0.23 | [−0.31, −0.15] | −0.18 | [−0.27, −0.10] | −0.03 | [−0.05, −0.01] | Partial mediation |
| SE-DO [Ind2] | −0.23 | [−0.31, −0.15] | −0.18 | [−0.27, −0.10] | −0.05 | [−0.09, −0.02] | Partial mediation |
| CO-EM | −0.45 | [−0.52, −0.38] | −0.45 | [−0.53, −0.37] | - | - | Direct relationship |
| CO-DO | −0.46 | [−0.54, −0.37] | −0.42 | [−0.51, −0.33] | −0.04 * | [−0.08, 0.00] | Direct relationship |
| SL-IM | 0.27 | [0.18, 0.35] | 0.27 | [0.18, 0.35] | - | - | Direct relationship |
| SL-DO | −0.04 | [−0.07, −0.02] | - | - | −0.04 | [−0.07, −0.02] | Partial mediation |
| EM-DO | 0.08 * | [−0.01, 0.17] | 0.08 * | [−0.01, 0.17] | - | - | No relationship |
| IM-DO | −0.14 | [−0.22, −0.06] | −0.14 | [−0.22, −0.06] | - | - | Direct relationship |

*Note.* SL = self-learning; EM = extrinsic motivation; IM = intrinsic motivation; CO = connectedness; SE = self-efficacy; DO = dropout; * = relationship not significant; Ind1 = indirect path SE-SL-IM-DO; Ind2 = indirect path SE-IM-DO.

### 3.5. Multi-Group Analysis

Using a multi-group analysis approach, we tested our structural model separately for different freshmen groups, carrying significant differences, as outlined in Table 5.

For the analysis of relationship status, we focused on the "single" and "engaged" groups, which comprised most of the sample. While most relationship statuses showed significant associations with dropout rates, we observed a significant path linking EM-DO only for engaged freshmen (β = 0.21, SE = 0.04, $p < 0.001$), and a significant path linking SE-DO only for single freshmen (β = −0.28, SE = 0.09, $p < 0.001$).

For the predictor of the parent's (father) education level, we compared three groups: "secondary school", "high school", and "college". We observed differences for the CO-EM path, which was significant for all groups but to a significantly lesser extent for freshmen whose parental education was at secondary school level (β = 0.38, SE = 0.08, $p < 0.001$). The same pattern was observed for the SE-IM and SE-DO paths, where we observed no significant associations for freshmen whose parents' education level was at secondary school.

When conducting analysis on the incoming school, the results revealed that freshmen coming from a vocational institute did not show significance for the SE-SL, SE-IM, SE-DO, and IM-DO paths. Freshmen coming from a high school showed significance for EM-DO (β = 0.11, SE = 0.03, $p = 0.027$), whereas a significant difference was observed for the SL-IM path, between freshmen coming from a technical institute (β = 0.21, SE = 0.13, $p = 0.003$) and those coming from a vocational institute (β = 0.49, SE = 0.49, $p = 0.002$).

Furthermore, we analyzed groups based on the area of study. Law freshmen did not show significance for three paths in the model, including SE-SL, SE-IM, and SL-IM, but they showed a significant association for IM-DO (β = −0.32, SE = 0.23, $p = 0.030$). Engineering freshmen were the only ones presenting significance for the EM-DO path (β = 0.19, SE = 0.04, $p = 0.005$). An interesting finding was that the CO-EM relationship was significantly different when comparing freshmen of medicine (β = −0.50, SE = 0.12, $p < 0.001$) to those of economics (β = −0.32, SE = 0.12, $p < 0.001$) and law (β = −0.42, SE = 0.10, $p = 0.024$), as well as when comparing freshmen of engineering (β = −0.47,

SE = 0.11, $p < 0.001$) to those of law. Finally, the SE DO relationship was significant only for freshmen of medicine ($\beta = -0.21$, SE = 0.10, $p = 0.001$) and freshmen of economics ($\beta = -0.21$, SE = 0.14, $p = 0.010$).

Regarding the proportion of class attendance, we found that freshmen attending less than 70% of classes did not show significance for the SE-IM and EM-DO paths, whereas freshmen attending more classes did not show significance for the IM-DO path.

**Table 5.** Multi-group analysis.

| | n | SE-SL | CO-EM | SE-IM | SL-IM | CO-DO | IM-DO | EM-DO | SE-DO |
|---|---|---|---|---|---|---|---|---|---|
| *Gender* | | ns | ns | ns | ns | ns | ns | ns | ns |
| Male (1) | 284 | 0.45 *** (0.10) | −0.43 *** (0.12) | 0.23 ** (0.16) | 0.28 *** (0.16) | −0.42 *** (0.07) | −0.15 * (0.05) | 0.08 ns (0.03) | −0.17 * (0.11) |
| Female (2) | 501 | 0.42 *** (0.06) | −0.46 *** (0.07) | 0.25 *** (0.14) | 0.23 *** (0.16) | −0.41 *** (0.05) | −0.14 *** (0.03) | 0.08 ns (0.03) | −0.19 *** (0.08) |
| *Relationship status* | | ns | ns | ns | ns | ns | ns | (1 vs. 2) | (1 vs. 2) |
| Single (1) | 451 | 0.38 *** (0.07) | −0.48 *** (0.08) | 0.27 *** (0.15) | 0.28 *** (0.13) | −0.37 *** (0.05) | −0.10 * (0.03) | 0.02 ns (0.03) | −0.28 *** (0.09) |
| Engaged (1) | 311 | 0.51 *** (0.08) | −0.45 *** (0.11) | 0.19 * (0.16) | 0.27 *** (0.19) | −0.46 *** (0.07) | −0.11 * (0.04) | 0.21 *** (0.04) | −0.09 ns (0.09) |
| *Parents' education level* | | ns | (1 vs. 2, 3) | (1 vs. 2, 3) | ns | ns | ns | ns | (1 vs. 2, 3) |
| Secondary school (1) | 276 | 0.44 *** (0.09) | −0.38 *** (0.08) | 0.12 ns (0.17) | 0.37 *** (0.17) | −0.53 *** (0.05) | −0.24 *** (0.04) | 0.02 ns (0.04) | −0.11 ns (0.09) |
| High school (2) | 346 | 0.39 *** (0.07) | −0.48 *** (0.10) | 0.29 *** (0.15) | 0.21 ** (0.15) | −0.46 *** (0.08) | −0.12 * (0.04) | 0.08 ns (0.03) | −0.17 ** (0.10) |
| University (3) | 113 | 0.39 *** (0.07) | −0.48 *** (0.10) | 0.29 *** (0.15) | 0.21 ** (0.15) | −0.46 *** (0.08) | −0.12 * (0.04) | 0.08 ns (0.03) | −0.17 ** (0.10) |
| *Incoming school* | | (3 vs. 1, 2) | ns | (3 vs. 1, 2) | (2 vs. 3) | ns | (3 vs. 1, 2) | (1 vs. 2, 3) | (3 vs. 1, 2) |
| High school (1) | 437 | 0.46 *** (0.07) | −0.48 *** (0.09) | 0.20 *** (0.14) | 0.28 *** (0.16) | −0.41 *** (0.06) | −0.11 * (0.03) | 0.11 * (0.03) | −0.18 * (0.09) |
| Technical institute (2) | 290 | 0.38 *** (0.08) | −0.40 *** (0.09) | 0.29 *** (0.16) | 0.21 ** (0.13) | −0.44 *** (0.06) | −0.17 ** (0.04) | 0.05 ns (0.03) | −0.14 * (0.10) |
| Vocational institute (3) | 63 | 0.14 ns (0.17) | −0.54 *** (0.23) | 0.26 ns (0.49) | 0.49 ** (0.49) | −0.42 * (0.13) | −0.16 ns (0.05) | 0.08 ns (0.07) | −0.25 ns (0.27) |
| *Area of study* | | (4 vs. 1, 2, 3) | (1 vs. 2, 4) (3 vs. 4) | (4 vs. 1, 2, 3) | (4 vs. 1, 2, 3) | ns | (4 vs. 1, 2, 3) | (3 vs. 1, 2, 4) | (1, 2 vs. 3, 4) |
| Medicine (1) | 337 | 0.40 *** (0.08) | −0.50 *** (0.12) | 0.29 *** (0.19) | 0.18 ** (0.18) | −0.39 *** (0.07) | −0.14 ** (0.03) | 0.04 ns (0.03) | −0.21 ** (0.10) |
| Economy (2) | 195 | 0.42 *** (0.13) | −0.32 *** (0.12) | 0.24 * (0.19) | 0.28 ** (0.14) | −0.38 *** (0.06) | −0.08 ns (0.06) | 0.08 ns (0.04) | −0.21 ** (0.14) |
| Engineering (3) | 205 | 0.44 *** (0.08) | −0.47 *** (0.11) | 0.19 * (0.16) | 0.31 ** (0.22) | −0.45 *** (0.07) | −0.05 ns (0.05) | 0.19 ** (0.04) | −0.15 ns (0.11) |
| Law (4) | 53 | 0.31 ns (0.14) | −0.42 * (0.10) | −0.01 ns (0.16) | 0.43 ns (0.35) | −0.62 ** (0.19) | −0.32 * (0.23) | −0.05 ns (0.21) | −0.02 ns (0.25) |
| *Class attendance* | | ns | ns | (1 vs. 2) | ns | ns | (1 vs. 2) | (1 vs. 2) | ns |
| Less than 70% (1) | 148 | 0.33 ** (0.13) | −0.38 *** (0.13) | 0.19 ns (0.18) | 0.20 * (0.14) | −0.43 *** (0.08) | −0.30 ** (0.08) | 0.03 ns (0.05) | −0.18 * (0.13) |
| More than 70% (2) | 642 | 0.43 *** (0.06) | −0.45 *** (0.08) | 0.21 *** (0.12) | 0.30 *** (0.14) | −0.40 *** (0.05) | −0.08 ns (0.02) | 0.11 ** (0.03) | −0.18 *** (0.07) |

*Note.* SL = self-learning; EM = extrinsic motivation; IM = intrinsic motivation; CO = connectedness; SE = self-efficacy; DO = dropout; ns = not significant; * = significant at 0.05; ** = significant at 0.01; *** = significant at <0.001. Values represent standardized estimates, with standard errors reported within parentheses.

## 4. Discussion

This study aimed at testing hypothesized relationships linking self-efficacy and connectedness to dropout intention, which were mediated by learning strategies and motivation. The results show that connectedness and self-efficacy significantly predict dropout individually, the latter also in joint action with identified mediators.

Self-efficacy was inversely related to dropout, also after controlling for the effects of learning strategies and of intrinsic motivation. Freshmen with a good sense of self-efficacy are, indeed, less prone to dropout, leveraging internal resources such as the ability to regulate and focus on study, while making more use of active support. This result replicates the past literature where low self-efficacy was associated with reduced belief in one's own skills and abilities [45].

We further observed that the relationship is stronger for singles (compared to engaged), higher education of father (college vs. secondary school), coming from high schools and technical institutes (compared to vocational), and being enrolled in medicine and economics areas (compared to engineering and law). These findings are not totally unexpected, considering past research. Take, for instance, incoming school. Students who attend

schools with lower academic performance or selectivity may be at a higher risk of academic dropout [46]. These students may be less prepared for the rigors of university coursework and may face more challenges in adapting to the academic environment. However, students might react differently to similar areas of study at the academic level, as certain fields of study have higher dropout rates than others. For example, STEM is associated with higher dropout rates compared to other fields [47]. This may be due to the challenging coursework and lack of support for students in these fields, and this is partly supported by our analysis. The supporting literature also exists when considering parents' education level, as it was reported that students whose parents have lower levels of education are at a higher risk of academic dropout [1]. This may be due to a lack of awareness of the importance of education, as well as a lack of support and resources at home. Conversely, past research showed that being in a romantic relationship is associated with a lower likelihood of academic dropout [48]. This may be due to the emotional and social support provided by a partner. Following these findings, we deem that it is crucial to provide targeted training interventions for freshmen to help them identify their strengths and weaknesses and develop their skills, focusing especially on the groups at risk identified in the model.

In our model, connectedness was also strongly and inversely related to the dropout intention. However, in this case, extrinsic motivation was not a good predictor for dropout; hence, only a direct relationship was established between connectedness and dropout. Compared to self-efficacy, a more homogenous picture emerged when testing this relationship in further analyses between groups, as we did not observe any significant difference. Interpreting the results, freshmen who are supported toward nurturing their connectedness to the academic community may be less likely to drop out. The awareness of belonging to a social group contributed toward the definition and cohesion of their personal and cultural identity. It is fundamental for freshmen to define the cultural boundaries within which their studies are located, which over time could constitute a protective factor against dropout. A university capable of creating a sense of belonging through its educational programs could have fewer students who drop out [49].

We further explored the effects of the mediators in the outcome. The results show that self-regulated knowledge has an important role in predicting dropout intention from self-efficacy, by acting as a joint mediator together with intrinsic motivation. The literature shows that a low level of self-regulated learning is related to a higher risk of dropout [50]. Further studies showed that self-regulated knowledge is a protective factor against dropout and is functional for developing optimal learning strategies and for fueling the perception of self-efficacy in a good vicious circle [51]. To increase self-regulated knowledge skills and reduce the risk of a possible dropout, it would be essential to integrate educational guidance services that support students in working on their study method.

Intrinsic motivation was also shown to be significantly and inversely related to dropout. This construct represents the highest level of self-determination, which implies the deployment of autonomous regulation strategies. This is associated with feelings of competence and autonomy, which are displayed by students who choose university on the basis of their interest and pleasure in studying a particular subject and discovering new knowledge within a specific domain, as well as on the sense of satisfaction they experience when studying [52]. This result confirms other studies that found that self-determined motivation to attend university is associated with greater academic achievements and student retention [21,53].

Follow-up analyses between groups carried further evidence, showing the effect of the mediator to be stronger for students coming from high schools and technical institutes, enrolled in law, and attending more than 70% of class. Low attendance rates were found to be a predictor of academic dropout [32]. Students who miss a significant number of classes may fall behind in coursework and may not have access to important information and resources provided in class.

This study has a few limitations, mainly due to the generalizability of the results, since data were gathered sampling only freshmen of a single university. Moreover, the

areas of study are quite limited with a very small proportion of law. In addition, the use of web-based surveys may exclude non-digitalized students, who may have different socio-economic characteristics and lifestyles compared to those who have Internet access. Therefore, conducting a survey exclusively online may result in under-representing certain social groups and introducing significant bias.

## 5. Conclusions

Universities should invest in training and orientation programs to enhance self-efficacy, motivation, and university connectedness among freshmen in order to improve their university experience. A better university experience has a positive impact not only on academic performance but also on student well-being, reducing mental distress in college students. It should be noted that college students are in a crucial period of transition from late-adolescence to adulthood, where they have to deal with important stressful tasks.

Accompanying actions are needed at different levels (e.g., peer education, mentoring, tutoring) in order to prevent dropout and promote empowerment, supported also by the development of transversal skills [54]. The results confirm the importance of improving students' ability to regulate and focus on academic support, new study methods, and better self-organization. This is especially important for freshmen, because the first year of university is the most critical. The academic environment can trigger a wide variety of emotions. It has been shown that some of these emotions can have an effect on academic performance and be associated with the intention to drop out of studies; among these, anxiety seems to play an important role [55]. For this reason, it would be important to also support freshmen with psychological counseling, which helps them to manage emotions and stressors related to university. Only by considering all these factors will it be possible to promote students' well-being and their academic performance, reducing dropout.

**Author Contributions:** Conceptualization, C.B. and A.G.; methodology, C.B., A.G., G.S., S.B. and G.R.; formal analysis, H.C. and C.B.; investigation, C.B. and G.S.; resources, A.G.; data curation, G.S. and H.C.; writing—original draft preparation, C.B. and H.C.; writing—review and editing, C.B., H.C., S.B. and A.G.; supervision, C.B. and A.G.; project administration, A.G.; funding acquisition, A.G. All authors have read and agreed to the published version of the manuscript.

**Funding:** This work received a specific grant from the Italian Ministry of University and Research for tutoring orientation actions, as well as recovery and inclusion actions, also with reference to students with disabilities and specific learning disabilities (D.M. n. 752, 30 June 2021).

**Institutional Review Board Statement:** Organizational ethics approvals were obtained from the Board of Directors of Brescia University (approved with provision no. 330 on 22 November 2021). This study was performed in accordance with the ethical standards as laid down in the 1964 Declaration of Helsinki and its later amendments. Students were informed that their participation was confidential, anonymous, not compulsory, and that their personal data would be respected.

**Informed Consent Statement:** Informed consent was obtained from all subjects involved in this study.

**Data Availability Statement:** The data that support the findings of this study are available on request from the corresponding author.

**Conflicts of Interest:** The authors declare no conflicts of interest.

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
