# Peer review of "The Role of Self-Efficacy, Motivation, and Connectedness in Dropout Intention in a Sample of Italian College Students"

_education, doi:10.3390/educsci14010067_

Round 1

Reviewer 1 Report

Comments and Suggestions for Authors

Thank you for the opportunity to review, The Role of Self-Efficacy, Motivation and Connectedness in Dropout Intention in a Sample of Italian College Students. This is an important topic, particularly within an Italian context, where the authors report the highest levels of college dropouts occur within Europe. The authors aim was to evaluate potential mediators, motivation and cognitive strategies for learning, within relationships of self-efficacy and connectedness as they relate to drop-out intention. Participants included 790 Italian college freshmen who responded via a web-survey including the Academic Motivation Scale, Perceived School Self-Efficacy Scale, University Connectedness Scale, Self-Regulated Knowledge Scale-University, and five questions regarding intention to dropout. A multiple regression analysis, confirmatory factor analysis, and structural equation modeling were utilized. Findings suggested that self-efficacy is the strongest predictor of dropout intentions, with connectedness the next strongest predictor, with both mediated by learning strategies and motivation.

The manuscript addresses an important topic, the evaluation is comprehensive, and the paper is well written. Despite these strengths, major areas for recommended revision are outlined below:

·         pp. 1-2 The manuscript would benefit from increased clarity and parenthetical support to clarify the constructs selected for evaluation and their relationship to the others. It is unclear how the literature base informed the selection of self-efficacy, connectedness, and self-regulated learning. The constructs selected appear somewhat disjointed given the current introduction. Further operationalization with parenthetical support for the constructs is also suggested.

·         Pp. 1-2 The emphasis on intrinsic vs. extrinsic motivation is also unclear, which is also related to the earlier comment.

·         p. 2 The recommendation above would support greater clarity of the study’s purpose.

Minor areas of weakness are noted below:

·         p. 4 Generalization – the areas of study are quite limited (i.e., medicine, engineering, and economics, with a very small proportion of law) and race/ethnicity data were not collected/reported.

·         p. 11 line 285 – Template wording is included, “This section…”

Reviewer 2 Report

Comments and Suggestions for Authors

The paper conducts a thorough and robust examination of how self-efficacy, motivation, and connectedness affect dropout intention among Italian college students. The paper employs a sizable and diverse sample, valid and reliable instruments, and suitable statistical techniques. The paper also suggests useful recommendations for universities to improve the academic outcomes and persistence of their students, by developing and applying interventions that foster self-efficacy, motivation, and connectedness. This paper makes an important contribution to the research on student dropout and retention, and I endorse the paper for publication.

Author Response

Thank you very much for taking the time to review this manuscript. Thank you for appreciating our article and authorizing its publications.

Best regards

Reviewer 3 Report

Comments and Suggestions for Authors

This study collected 790 data and used multiple regression analysis, confirmatory factor analysis and structural equation modeling to analyze the data.

This article has the following issues that need clarification:

1. This article uses an 11-point Likert scale for the Academic Motivation Scale (AMS), a 5-point Likert scale for the Perceived School Self-Efficacy Scale (PSSES), and a 7-point Likert scale for the University Connectedness Scale (UCS). The Self-Regulated Knowledge Scale-University (SRKS-U) was scored on a 5-point Likert scale, and the dropout intention of freshmen was scored on a 5-point Likert scale.

If the scoring range of the scale used is inconsistent, it should be explained why this is done and how to make the scoring standards of all aspects consistent during statistical analysis.

2. The correlation literature between the dimensions proposed in the introduction of Chapter 1 is incomplete, which will make the overall research model unable to meet the requirements of academic validity.

3. The conclusion of this article is too simple and some more content should be added.

Author Response

Please find the detailed responses below and the corresponding corrections highlighted in yellow in the re-submitted file.

Thank you!

Reviewer 4 Report

Comments and Suggestions for Authors

The article employs a variety of analytical methods, providing a comprehensive insight into the relationships between the studied variables. Additionally, the research addresses a significant social concern, college dropout, offering valuable insights for educational institutions and policymakers to enhance student retention. Thus, the article delivers a robust analysis of psychosocial variables related to college dropout, emphasizing the importance of self-efficacy and university connection. While it represents a valuable contribution, limited generalization and the potential inclusion of additional variables are areas that could be improved in future research. Overall, considering its positive aspects, the article makes a valuable contribution to the understanding and approach to college dropout.

Author Response

Thank you very much for taking the time to review this manuscript. Thank you for appreciating our article. We further underlined the limited generalization of our study in the limitations section. 

Best regards